# Enzyme-Assisted Mechanical Peeling of Cassava Tubers

**Ziba Barati ***, **Sajid Latif, Sebastian Romuli and Joachim Müller**

Institute of Agricultural Engineering (440e), Tropics and Subtropics Group, University of Hohenheim, 70599 Stuttgart, Germany; s.latif@uni-hohenheim.de (S.L.); Sebastian_Romuli@uni-hohenheim.de (S.R.); Joachim.mueller@uni-hohenheim.de (J.M.)
* Correspondence: Barati@uni-hohenheim.de; Tel.: +49-711-459-24704; Fax: +49-711-459-23298

**Abstract:** In this study, the effect of enzymatic pre-treatment and the size of cassava tubers on mechanical peeling was examined. Cassava tubers were sorted based on their mass as small, medium and large. Viscozyme® L and an abrasive cassava peeling machine was used for the enzymatic pre-treatment and the mechanical peeling, respectively. Response surface methodology (RSM) was used to investigate the effect of the enzyme dose (0.5–1.9 mL $g^{-1}$), incubation time (1.5–6 h), peeling time (1.5–4.5 min) and size of the tubers (small, medium and large) on the peeling process. Peeled surface area (PSA) and peel loss (PL) were measured as main responses in RSM. Results showed that the PSA and PL were significantly ($p < 0.05$) influenced by the enzyme dose, incubation time and peeling time. The size of tubers only had a significant impact on the PSA. The optimum operating conditions for different sizes of tubers were found and validated. Under optimum conditions, the PSA of the large tubers (89.52%) was significantly higher than the PSA of the medium and small tubers ($p < 0.05$). Application of enzymatic pre-treatment can improve the mechanical peeling process especially for larger cassava tubers.

**Keywords:** enzymatic treatment; mechanical peeling; peeling efficiency; response surface method; size of tubers

---

## 1. Introduction

Cassava is one of the most important crops in the world and is mostly consumed as food, feed and is used as industrial raw material [1,2]. It is a resilient crop, which can grow under poor conditions such as drought and low nutrient content [3] and provides a staple food for around one billion people [1]. Due to the climate change, the importance of resilient crops such as cassava to secure the food supply has gained more attention [4,5]. Cassava should be processed after harvesting due to its short shelf life [6]. Peeling, as the first step for all cassava products such as flour, starch and gari, is one of the bottlenecks in processing due to the different shapes and sizes of the tubers [7,8]. Because manual peeling is labor intensive [9], it is essential to mechanize the enzymatic cassava peeling process. Regarding the mechanical peeling of cassava tubers, the cassava peeling machines available on the market, which were developed in Nigeria, Brazil and China, are not efficient enough [10–13] and cause high flesh losses of about 25–40% [14]. Therefore, finding a way to increase the efficiency of mechanical peeling is very important. The sorting and grading of crops, especially for those with an irregular shape and size, is a common post-harvest practice prior to mechanical processing [15]. In addition, previous studies have emphasized the importance of sizing to improve the peeling efficiency of cassava tubers [16,17]. However, the sorting of cassava tubers for later processing operations has scarcely been studied up to now. Different peeling methods, including manual, mechanical, chemical and thermal peeling, have been studied and evaluated in the literature [7]. Enzymatic peeling, as a novel

concept, has been practiced on different fruits, vegetables [18–24] and also tubers like potato and sweet potato [25,26]. Enzyme application is considered to be a green technology, which can be a solution for sustainable food processing in the future [27]. The use of enzymes was found to be a promising pre-treatment to improve the manual peeling of cassava [28].

Besides the above-mentioned facts, the amount of cassava peel waste generated from cassava processing is high and not sufficiently utilized [29]. Cassava peels are a lignocellulose biomass, which can be used for ethanol production as a source of renewable energy [30]. However, the matrix of lignin and hemicellulose has a barrier function for the digestion of the peels [30]. Hydrolysis is one of the essential pre-treatments for cassava peels to degrade the peels to simple sugars, which can be further converted to ethanol [29,30]. The use of enzymes for peeling can facilitate the hydrolysis of the peels for their later application in ethanol production. Moreover, the cassava peels contain toxic compounds of cyanogenic glucosides such as linamarin and lotaustralin [31], which have a negative impact on the environment. Use of enzymes to hydrolyze the peels can catalyze the detoxification of the cassava peels by increasing the contact of linamarin and linamarase [32].

To the best of our knowledge, no study has reported the use of enzyme treatment for the mechanical peeling of cassava tubers. Therefore, the main objective of the present study is to investigate the effect of enzyme treatment and the grading of tubers based on their mass on the efficiency of the mechanical peeling process.

## 2. Results and Discussions

### 2.1. The Effect of Different Incision Patterns for Enzyme Infusion on the Peeling Process of Cassava Tubers

The comparison of peeled surface area (PSA) and peel loss (PL) of cassava tubers by different incision patterns after the mechanical peeling with enzyme treatment is presented in Table 1. The PSA and PL of cassava tubers treated with enzyme solution were higher than those of cassava tubers treated with blank solution and control. This finding was in accordance to our previous work [28]. Additionally, in preparatory experiments (data not shown), the efficiency of the enzyme was investigated on tubers without applying incisions on their surface, resulting in a PSA of 24.1 ± 0.04% and PL of 3.6 ± 0.6%. It was observed that the outer layer of the cassava peels could prevent the infusion of the enzyme. The lignin content of the corky epidermis (outer layer of the cassava peels) has a negative impact on enzymatic hydrolysis [33]. This result illustrates the importance of incisions in the peel of cassava tubers to allow enzymes to penetrate the peel. The enzyme treatment significantly increased the PSA and the PL of the cassava tubers ($p < 0.05$). Furthermore, it was observed that the cassava tubers with meridian incisions had the highest PSA and PL compared to those with perforation and spiral incision (Table 1). Therefore, meridian incision was chosen for the enzymatic treatment in the present study. However, there was no considerable difference amongst PSA and PL of tubers with different incision patterns for the blank solution and control.

**Table 1.** Peeled surface area (PSA) and peel loss (PL) of cassava tubers by different incision patterns (spiral, meridian and perforation) after mechanical peeling.

| Treatment | Incision Pattern | PSA (%) | PL (%) |
|---|---|---|---|
| Enzyme solution | Spiral | 67.3 [Ab] ± 7.2 | 6.5 [Ac] ± 1.7 |
| | Meridian | 91.3 [Aa] ± 5.7 | 20.8 [Aa] ± 2.3 |
| | Perforation | 80.7 [Aab] ± 5.3 | 12.8 [Ab] ± 0.7 |
| Blank solution | Spiral | 58.3 [Bc] ± 5.4 | 8.6 [Bc] ± 2.7 |
| | Meridian | 53.8 [Bc] ± 3.3 | 7.4 [Bc] ± 0.7 |
| | Perforation | 54.6 [Bc] ± 3.5 | 9.2 [Bc] ± 0.3 |
| Control | Spiral | 31.7 [Cd] ± 4.5 | 4.0 [Cd] ± 0.7 |
| | Meridian | 28.0 [Cd] ± 3.0 | 3.7 [Cd] ± 1.2 |
| | Perforation | 25.0 [Cd] ± 6.6 | 3.6 [Cd] ± 0.6 |

Data are presented as mean ± SD. The differences among mean values calculated using ANOVA are indicated by different letters in a column ($p < 0.05$); capital letters (A–C) indicate differences among treatment and lower case letters (a–d) indicate differences among incision.

Several researchers have studied the effect of different incision patterns on the infusion of enzymes in the peels of citrus fruits for enzymatic peeling [34,35]. It was reported, that incision for supporting of the infusion of enzymes is an essential step for enzymatic peeling.

### 2.2. The Effect of Enzyme Dose, Incubation Time, Peeling Time and Size of Tubers on the Peeling Process of Cassava Tubers

The design matrix of response surface methodology (RSM) with experimental and predicted values of PSA and PL is presented in Table 2. The PSA and PL ranged from 55% to 96% and from 6.2% to 35.6%, respectively under various operating conditions (enzyme dose, incubation time, peeling time) and size of tubers. The average PSA and PL of the cassava tubers was 80.92% and 21.30%, respectively. The highest PSA (96%) was achieved with an enzyme dose of 1.9 mL g$^{-1}$ of the sample, incubation time of 4 h and peeling time of 4.5 min for small cassava tubers. However, PL was 35.6%, indicating a high loss of flesh.

**Table 2.** Design matrix of response surface methodology (RSM) and experimental (EXP) and predicted (Pred) values for peeled surface area (PSA) and peel loss (PL) of enzyme treated cassava tubers after mechanical peeling.

| | Independent Variables | | | | Responses | | | |
|---|---|---|---|---|---|---|---|---|
| Run | Enzyme Dose (mL g$^{-1}$) | Incubation Time (h) | Peeling Time (min) | Size | PSA (%) | | PL (%) | |
| | $X_i$ | $X_j$ | $X_k$ | $X_f$ | Exp | Pre | Exp | Pred |
| 1 | 1.9 | 4 | 4.5 | Small | 96.0 | 97.6 | 35.2 | 34.3 |
| 2 | 1.2 | 4 | 3 | Large | 80.7 | 81.6 | 19.0 | 19.9 |
| 3 | 1.2 | 6.5 | 4.5 | Large | 94.0 | 93.6 | 33.7 | 34.2 |
| 4 | 1.2 | 6.5 | 1.5 | Large | 73.0 | 72.7 | 12.8 | 12.7 |
| 5 | 0.5 | 4 | 1.5 | Large | 69.0 | 66.2 | 6.2 | 5.7 |
| 6 | 1.9 | 1.5 | 3 | Medium | 79.8 | 80.5 | 24.4 | 24.4 |
| 7 | 1.2 | 4 | 3 | Large | 83.5 | 81.6 | 20.5 | 19.9 |
| 8 | 1.2 | 1.5 | 1.5 | Large | 69.0 | 70.2 | 9.3 | 11.3 |
| 9 | 1.2 | 1.5 | 4.5 | Large | 91.8 | 91.1 | 33.6 | 32.8 |
| 10 | 0.5 | 4 | 4.5 | Medium | 89.5 | 90.2 | 25.1 | 26.6 |
| 11 | 1.2 | 4 | 3 | Medium | 78.2 | 78.6 | 21.0 | 21.1 |
| 12 | 1.2 | 4 | 3 | Medium | 78.0 | 78.6 | 20.4 | 21.1 |
| 13 | 1.2 | 4 | 3 | Small | 85.6 | 84.3 | 23.2 | 23.1 |
| 14 | 1.9 | 1.5 | 3 | Small | 87.7 | 88.8 | 25.0 | 24.4 |
| 15 | 0.5 | 6.5 | 3 | Medium | 76.9 | 76.7 | 20.3 | 20.2 |
| 16 | 1.2 | 4 | 3 | Small | 81.5 | 84.3 | 19.5 | 23.1 |
| 17 | 1.2 | 1.5 | 1.5 | Small | 60.0 | 63.5 | 7.2 | 8.3 |
| 18 | 1.9 | 6.5 | 3 | Large | 85.4 | 86.1 | 27.0 | 26.9 |
| 19 | 0.5 | 4 | 4.5 | Large | 88.6 | 91.0 | 27.9 | 27.2 |
| 20 | 1.9 | 4 | 1.5 | Medium | 72.2 | 73.1 | 14.8 | 15.5 |
| 21 | 0.5 | 4 | 1.5 | Medium | 62.0 | 63.1 | 10.5 | 10.0 |
| 22 | 1.2 | 4 | 3 | Medium | 79.5 | 78.6 | 22.6 | 21.1 |
| 23 | 0.5 | 4 | 4.5 | Small | 91.4 | 89.9 | 25.7 | 26.2 |
| 24 | 0.5 | 1.5 | 3 | Large | 75.0 | 77.2 | 16.6 | 16.3 |
| 25 | 1.2 | 4 | 3 | Small | 83.0 | 84.3 | 22.1 | 23.1 |
| 26 | 1.2 | 4 | 3 | Large | 83.3 | 81.6 | 20.0 | 19.9 |
| 27 | 1.9 | 4 | 4.5 | Large | 95.0 | 93.7 | 35.6 | 36.5 |
| 28 | 1.2 | 6.5 | 1.5 | Small | 65.6 | 65.9 | 8.6 | 9.9 |
| 29 | 1.2 | 6.5 | 4.5 | Small | 94.0 | 95.0 | 30.6 | 29.2 |
| 30 | 1.9 | 4 | 1.5 | Small | 78.1 | 72.4 | 17.3 | 15.0 |
| 31 | 1.2 | 6.5 | 4.5 | Medium | 91.5 | 92.6 | 31.9 | 31.2 |
| 32 | 0.5 | 4 | 1.5 | Small | 55.0 | 57.0 | 7.0 | 6.9 |
| 33 | 1.2 | 4 | 3 | Medium | 78.9 | 78.6 | 21.5 | 21.1 |
| 34 | 0.5 | 6.5 | 3 | Large | 80.2 | 79.6 | 18.0 | 17.7 |
| 35 | 1.2 | 4 | 3 | Large | 82.5 | 81.6 | 19.4 | 19.9 |
| 36 | 1.2 | 4 | 3 | Small | 83.9 | 84.3 | 22.9 | 23.1 |
| 37 | 1.9 | 6.5 | 3 | Small | 90.5 | 91.3 | 25.0 | 26.0 |
| 38 | 1.2 | 4 | 3 | Small | 86.4 | 84.3 | 24.9 | 23.1 |

**Table 2.** *Cont.*

| Run | Enzyme Dose (mL g$^{-1}$) | Incubation Time (h) | Peeling Time (min) | Size | PSA (%) | | PL (%) | |
|---|---|---|---|---|---|---|---|---|
| | X$_i$ | X$_j$ | X$_k$ | X$_f$ | Exp | Pre | Exp | Pred |
| 39 | 1.2 | 1.5 | 1.5 | Medium | 68.2 | 66.9 | 13.3 | 13.2 |
| 40 | 1.2 | 1.5 | 4.5 | Medium | 90.0 | 90.2 | 30.1 | 29.9 |
| 41 | 1.9 | 4 | 4.5 | Medium | 94.5 | 92.6 | 32.6 | 32.1 |
| 42 | 0.5 | 1.5 | 3 | Small | 79.7 | 77.3 | 18.5 | 16.3 |
| 43 | 1.9 | 1.5 | 3 | Large | 83.7 | 83.7 | 26.6 | 25.5 |
| 44 | 1.2 | 4 | 3 | Large | 80.4 | 81.6 | 19.0 | 19.9 |
| 45 | 1.2 | 4 | 3 | Medium | 78.6 | 78.6 | 21.4 | 21.1 |
| 46 | 0.5 | 1.5 | 3 | Medium | 75.2 | 74.2 | 18.6 | 18.9 |
| 47 | 0.5 | 6.5 | 3 | Small | 80.0 | 79.7 | 18.9 | 17.9 |
| 48 | 1.9 | 4 | 1.5 | Large | 74.8 | 76.6 | 16.5 | 15.0 |
| 49 | 1.2 | 1.5 | 4.5 | Small | 93.7 | 92.5 | 25.9 | 27.6 |
| 50 | 1.2 | 6.5 | 1.5 | Medium | 70.0 | 69.3 | 14.6 | 14.6 |
| 51 | 1.9 | 6.5 | 3 | Medium | 81.9 | 82.9 | 24.8 | 25.7 |

The mean absolute percentage error (*MAPE*) of PSA and PL was 1.27% and 4.49%, respectively.

### 2.2.1. Effect on Peeled Surface Area (PSA)

After removing the statistically insignificant terms, the following mathematical equation for PSA of cassava tubers was obtained from the polynomial regression model:

$$Y_{PSA} = \beta_0 + \beta_i \cdot X_i + \beta_j \cdot X_j + \beta_k \cdot X_k + \beta_{ik} \cdot X_i \cdot X_k + \beta_{kk} \cdot X_k^2, \tag{1}$$

where $Y_{PSA}$ is the peeled surface area (%) after mechanical peeling with enzymatic treatment, $X_i$ is the enzyme dose (mL g$^{-1}$ of peels), $X_j$ is the incubation time of enzyme treatment (h) and $X_k$ is the peeling time (min). The regression coefficients ($\beta$) for the peeled surface area of small, medium and large cassava tubers are shown in Table 3.

**Table 3.** Regression coefficients ($\beta$) for peeled surface area (PSA) of small, medium and large cassava tubers after enzymatic treatment and mechanically peeling.

| Size | $\beta_0$ | $\beta_i$ | $\beta_j$ | $\beta_k$ | $\beta_{ik}$ | $\beta_{kk}$ |
|---|---|---|---|---|---|---|
| Small | 7.33143 | 21.45714 | 0.488333 | 28.39307 | −4.39762 | −2.23654 |
| Medium | 48.2256 | 8.16964 | 0.488333 | 6.10829 | −1.24048 | 0.523519 |
| Large | 54.70226 | 4.24464 | 0.488333 | 6.11364 | 0.138095 | 0.116358 |

Increasing the enzyme dose, incubation time and peeling time showed a positive effect on the PSA. The effect of individual variables and their interaction on the PSA are shown in Table 4. The goodness of the model was indicated by an $R^2$ and adjusted $R^2$ of 0.982 and 0.975, respectively and the *MAPE* was 1.27%. Enzyme dose, incubation time, peeling time and size of cassava tubers significantly ($p < 0.05$) influenced their PSA. The interaction terms of enzyme dose and peeling time, enzyme dose, tuber size and peeling time and size of tubers also significantly ($p < 0.05$) affected the PSA. A higher $p$ value for the lack of fit ($p = 0.4336$) showed the fitness of the model.

Figure 1 presents the surfaces plots for the peeled surface area as a function of the enzyme dose and peeling time, while keeping the incubation time constant, for different sizes of cassava tubers. The model was further verified with the normal probability plot for the externally studentized residuals. It was determined that most of the residuals were on a straight line (Figure S1a). This indicates the normal distribution of the data. Furthermore, the plot of residuals versus predicted values, as presented in Figure S1b, shows no clear pattern among the data, which suggests the absence of biases.

**Table 4.** ANOVA for reduced polynomial equation for the effect of enzymatic treatment, peeling time and size of tubers on the PSA of cassava tubers.

| Source | Sum of Squares | Degree of Freedom | Mean Square | F Value | p Value |
|---|---|---|---|---|---|
| Intercept | 4408.24 | 15.00 | 293.88 | 128.43 | <0.0001 |
| $X_i$—Enzyme dose | 394.31 | 1.00 | 394.31 | 172.32 | <0.0001 |
| $X_j$—Incubation time | 35.77 | 1.00 | 35.77 | 15.63 | 0.00 |
| $X_k$—Peeling time | 3582.42 | 1.00 | 3582.42 | 1565.58 | <0.0001 |
| $X_f$—Size | 83.57 | 2.00 | 41.78 | 18.26 | <0.0001 |
| $X_iX_k$ | 44.47 | 1.00 | 44.47 | 19.43 | <0.0001 |
| $X_iX_f$ | 36.07 | 2.00 | 18.04 | 7.88 | 0.00 |
| $X_kX_f$ | 70.53 | 2.00 | 35.26 | 15.41 | <0.0001 |
| $X_k{}^2$ | 18.22 | 1.00 | 18.22 | 7.96 | 0.01 |
| $X_iX_kX_f$ | 47.69 | 2.00 | 23.84 | 10.42 | 0.00 |
| $X_k{}^2X_f$ | 95.20 | 2.00 | 47.60 | 20.80 | <0.0001 |
| Residual | 80.09 | 35.00 | 2.29 | - | - |
| Lack of Fit | 54.64 | 23.00 | 2.38 | 1.12 | 0.43 |
| Pure Error | 25.45 | 12.00 | 2.12 | - | - |
| Correction total | 4488.32 | 50 | - | - | - |

$R^2$, 0.982; adjusted $R^2$, 0.975; and $p < 0.05$ indicates significance at the 95% level.

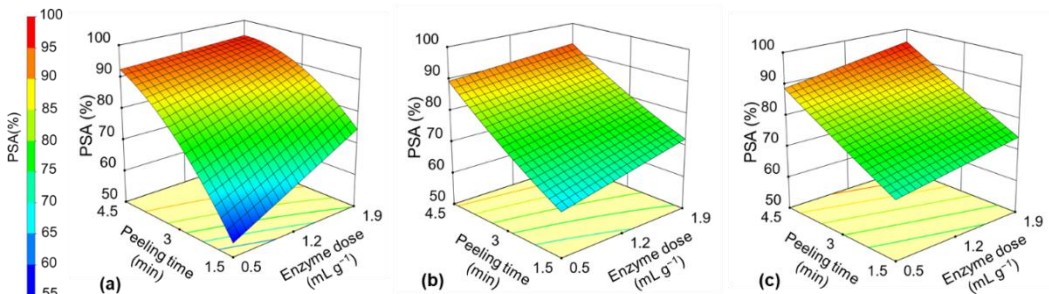

**Figure 1.** Peeled surface area (PSA) of cassava tubers after enzyme treatment and mechanical peeling vs. peeling time and the enzyme dose at an incubation time of 4 h; (**a**) small, (**b**) medium and (**c**) large size of cassava tubers.

### 2.2.2. Effect on Peel Loss (PL)

After removing the statistically insignificant terms, the following mathematical equation for the PL of cassava tubers was obtained from a polynomial regression model:

$$Y_{PL} = \beta_0 + \beta_i \cdot X_i + \beta_j \cdot X_j + \beta_k \cdot X_k + \beta_{jj} \cdot X_j^2 + \beta_{kk} \cdot X_k^2, \tag{2}$$

where $Y_{PL}$ is the peel loss (%) after mechanical peeling with enzymatic treatment, $X_i$ is the enzyme dose (mL g$^{-1}$ of peels), $X_j$ is the incubation time of enzyme treatment (h) and $X_k$ is the peeling time (min). The regression coefficients ($\beta$) for peel loss of small, medium and large cassava tubers are shown in Table 5.

**Table 5.** Regression coefficients ($\beta$) for peel loss (PL) of small, medium and large cassava tubers after enzymatic treatment and mechanically peeling.

| Size | $\beta_0$ | $\beta_i$ | $\beta_j$ | $\beta_k$ | $\beta_{ik}$ | $\beta_{kk}$ |
|---|---|---|---|---|---|---|
| Small | −19.0248 | 5.78393 | 2.73935 | 12.89886 | −0.30223 | −1.0762 |
| Medium | 1.3296 | 3.94464 | −1.24216 | 5.81237 | 0.187958 | −0.04567 |
| Large | −1.66721 | 6.6125 | −1.86041 | 4.05539 | 0.266926 | 0.518129 |

It was found that increasing the enzyme dose, incubation time and peeling time had a positive effect on the PL. The effect of individual variables and their interaction on the PL are shown in Table 6. The goodness of the model was indicated by an $R^2$ and adjusted $R^2$ of 0.979 and 0.968, respectively. The *MAPE* was 4.49%. Enzyme dose, incubation time and peeling time significantly ($p < 0.05$) influenced

the PL of cassava tubers. The interaction terms of enzyme dose and size of tubers and peeling time and size of tubers also significantly ($p < 0.05$) affected the PL. A higher $p$ value of the lack of fit ($p = 0.4509$) showed the fitness of the model.

Figure 2 presents the surface plots for the peel loss as a function of the enzyme dose and peeling time, while keeping the incubation time fixed at 4 h, for different sizes of cassava tubers. Unlike the peeled surface area, the size of the tubers did not significantly ($p > 0.05$) influence the PL of cassava tubers. The model was further analyzed with the normal probability plot for the externally studentized residuals. Similar to the PSA, the data was normally distributed (Figure S1c) and no clear patterns were identified among the data (Figure S1d).

**Table 6.** ANOVA for reduced polynomial equation for the effect of enzymatic treatment, peeling time and size of tubers on the peel loss of cassava tubers.

| Source | Sum of Squares | Degree of Freedom | Mean Square | F Value | *p* Value |
|---|---|---|---|---|---|
| Intercept | 2668.42 | 17.00 | 156.97 | 88.47 | <0.0001 |
| $X_i$—Enzyme dose | 348.92 | 1.00 | 348.92 | 196.66 | <0.0001 |
| $X_j$—Incubation time | 12.27 | 1.00 | 12.27 | 6.92 | 0.01 |
| $X_k$—Peeling time | 2198.99 | 1.00 | 2198.99 | 1239.39 | <0.0001 |
| $X_f$—Size | 3.24 | 2.00 | 1.62 | 0.91 | 0.41 |
| $X_i X_f$ | 14.62 | 2.00 | 7.31 | 4.12 | 0.03 |
| $X_j X_f$ | 0.10 | 2.00 | 0.05 | 0.03 | 0.97 |
| $X_k X_f$ | 23.89 | 2.00 | 11.94 | 6.73 | 0.00 |
| $X_j^2$ | 1.28 | 1.00 | 1.28 | 0.72 | 0.40 |
| $X_k^2$ | 2.60 | 1.00 | 2.60 | 1.46 | 0.23 |
| $X_j^2 X_f$ | 31.36 | 2.00 | 15.68 | 8.84 | 0.00 |
| $X_k^2 X_f$ | 27.94 | 2.00 | 13.97 | 7.87 | 0.00 |
| Residual | 58.55 | 33.00 | 1.77 | - | - |
| Lack of Fit | 38.45 | 21.00 | 1.83 | 1.09 | 0.45 |
| Pure Error | 20.10 | 12.00 | 1.68 | - | - |
| Correction total | 2726.97 | 50.00 | - | - | - |

$R^2$, 0.979; adjusted $R^2$, 0.968; and $p < 0.05$ indicates significance at the 95% level.

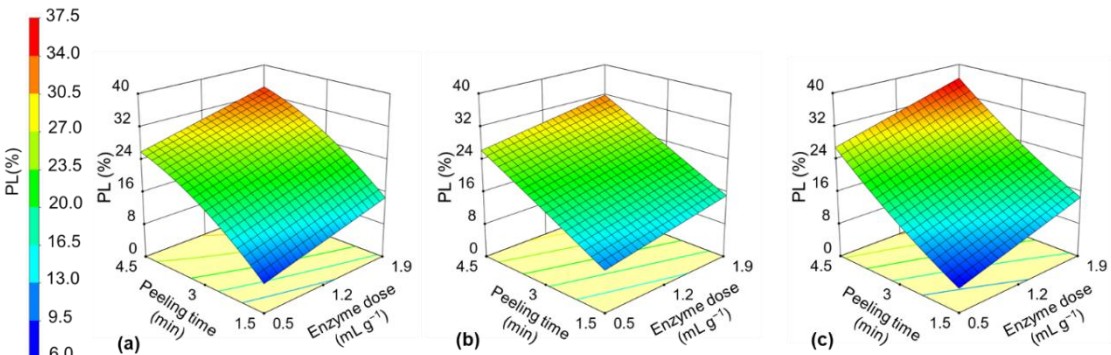

**Figure 2.** Peel loss (PL) of cassava tubers after enzyme treatment and mechanical peeling vs. peeling time and enzyme dose at incubation time of 4 h; (**a**) small, (**b**) medium and (**c**) large size of cassava tubers.

### 2.2.3. Optimal Operating Condition

Based on the model, an optimum peeling process could be achieved for different sizes of cassava tubers. The optimum operating conditions along with predicted and measured values of PSA and PL for different sizes of tubers are presented in Table 7.

Under optimum conditions, the measured PSA for small, medium and large cassava tubers was 82.03%, 80.90% and 89.52%, respectively. The PSA of large tubers was significantly higher than the PSA of small and medium tubers at $p < 0.05$. The reason could be that the estimated optimal concentration of the enzyme applied on larger tubers was higher than the concentration of the enzyme applied on small and medium sized tubers. This result is in accordance with the findings in the study of Jimoh and Olukunle [9], who reported that the peeling efficiency was higher for larger tubers than for the

other sizes. In the present study, the measured PL was 22.49%, 22.89% and 24.61% for small, medium and large tubers, under optimal conditions. There were no significant differences among the PL values of the different tuber sizes at $p < 0.05$.

**Table 7.** Optimum operating conditions in terms of enzyme dose, incubation time and peeling time along with the measured (Exp) and predicted (Pred) values of the peeled surface area (PSA) and peel loss (PL) of small, medium and large size cassava tubers.

| Size | Enzyme Dose (mL g$^{-1}$) | Incubation Time (h) | Peeling Time (min) | PSA (%) | | PL (%) | |
|---|---|---|---|---|---|---|---|
| | | | | Pred | Exp | Pred | Exp |
| Small | 0.8 | 4.0 | 3.0 | 81.87 | 82.03 [a] | 21.05 | 22.49 [a] |
| Medium | 0.5 | 2.9 | 3.5 | 79.76 | 80.90 [a] | 20.53 | 22.89 [a] |
| Large | 1.3 | 2.8 | 3.5 | 85.44 | 89.52 [b] | 24.28 | 24.61 [a] |

Different letters in a column show the significant difference among mean values ($p < 0.05$).

The peeling efficiency of enzyme-treated cassava tubers, which was determined in terms of PSA and PL in this study, was higher and more acceptable compared to other studies, which have used an abrasive peeling machine for peeling cassava tubers without any pre-treatments [7,9,36].

The findings of this study are in agreement with other studies applying enzymes for the peeling process of oranges, apricots, nectarines, peaches and grapefruits [18,20,23]. Enzyme application would facilitate the peeling process by hydrolyzing the peels. The use of enzymes would improve the peeling process by increasing the yield, reducing the losses and the peeling time compared to other peeling methods [37,38]. Additionally, the enzyme application could improve the quality of the products compared to other peeling methods in terms of appearance, texture, nutrients, and shelf-life [38–40]. However, the use of enzymes could increase the cost of the peeling process. Therefore, in order to reduce the cost, the reuse of enzymes should be further studied. Several studies have focused on reusing enzymes for the enzymatic peeling process of fruits [23,41]. It was found that ultrafiltration can be used to concentrate, purify and recover the enzyme solution and in some studies ultrafiltration has been successfully used to reuse the enzyme in the peeling process of lemons and grapefruit [23,41].

### 2.3. Correlation of Peel Loss with Crude Fiber, Starch Content and Color of the Peel Residue

A correlation was established between the crude fiber content (CF) of the peel residue and the PL through a linear model with an $R^2$ of 0.907 and a *MAPE* of 0.105%, which indicated a good fitting (Figure 3a, Table 8). As expected, increasing the PL resulted in a reduction of CF because more flesh with low crude fiber content is removed from the tubers, which dilutes the crude fiber content of the peel residue.

Contrary to the crude fiber content, the starch content (SC) of the peel residue increased with increasing PL, because the proportion of the flesh with a high starch content increases. A correlation between the starch content of the peel residue and PL resulted in a high $R^2$ of 0.969 and low *MAPE* of 0.026% (Figure 3b, Table 8).

The higher proportion of flesh in the peel at higher PL was also evident in the lightness (L) of the peel residue. A correlation between the L-value and PL, showed an $R^2$ of 0.934 and a *MAPE* of 0.017% (Figure 3c, Table 8). With increasing PL, the white color of the flesh increasingly dominates over the dark color of the peel in the peel residue. Based on this result an optical method for continuous in-line measurement of the peeling efficiency during mechanical peeling of cassava tubers could be developed.

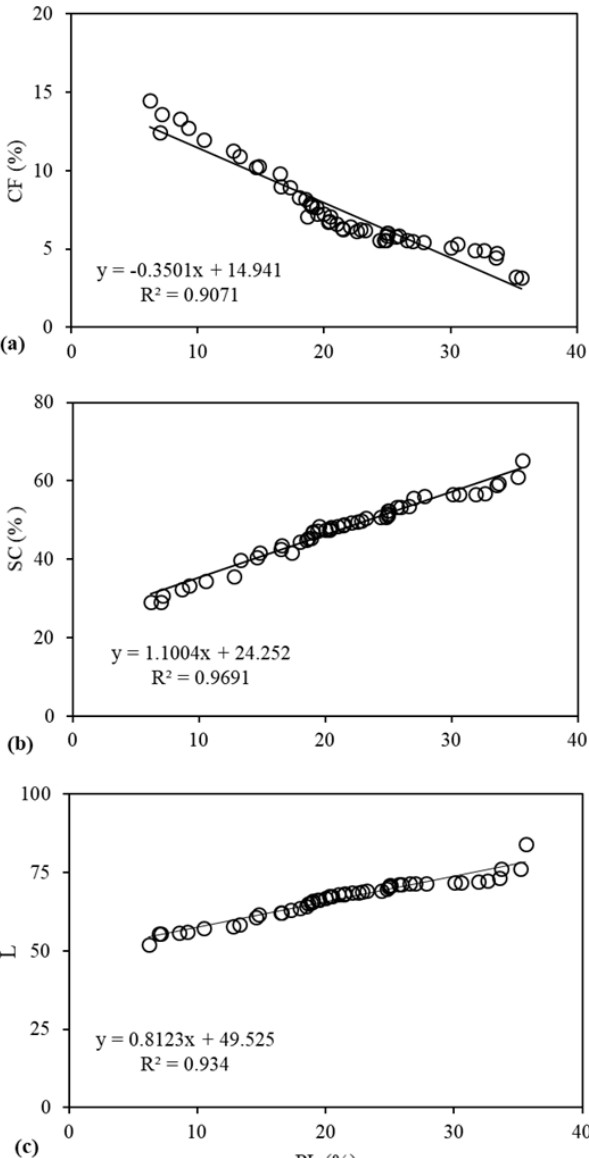

**Figure 3.** (**a**) Correlation of the crude fiber content (CF) of the peel residue with peel loss (PL), (**b**) correlation of the starch content (SC) of the peel residue with the peel loss (PL) and (**c**) correlation of the lightness (L) of the peel residue with the peel loss (PL) after mechanical peeling of enzyme treated cassava tubers.

**Table 8.** Linear mathematical models of crude fiber (CF), starch content (SC) and lightness (L) values of peel residues with peel loss (PL) after mechanical peeling of enzymatically treated cassava tubers.

| Mathematical Model | $R^2$ | *MAPE* (%) | Equation |
|---|---|---|---|
| CF = −0.3501·PL + 14.941 | 0.907 | 0.105 | (1) |
| SC = 1.1004·PL + 24.252 | 0.969 | 0.026 | (2) |
| L = 0.8123·PL + 49.525 | 0.934 | 0.017 | (3) |

## 3. Materials and Methods

### 3.1. Materials

#### 3.1.1. Plant Material

Cassava tubers imported from Costa Rica were purchased from a local market in Stuttgart. The tubers were classified and sorted based on their mass as small (350–550 g), medium (550–750 g) and large (750–1000 g). After sorting, the cassava tubers were stored in a refrigerator (7 °C) for the experiment.

#### 3.1.2. Enzyme

In line with our previous work [28], Viscozyme® L was used with hemicellulase and cellulase activities for the enzyme treatment in the present study. The enzyme was purchased from Univar GmbH, Essen, Germany.

### 3.2. Experimental Procedure

#### 3.2.1. Cassava Peeling Machine

An abrasive peeling machine was used in this study as described in detail in our previous work [42].

#### 3.2.2. Pre-Test on Peel Incision

Incisions in the peel are expected to facilitate enzyme infusion in enzymatic peeling of fruits and vegetables [22,35]. To identify a suitable incision method for cassava tubers, different incision patterns were investigated in a pre-test, namely, spiral incision, meridian incision and punctiform perforation (Figure 4). For spiral and meridian incisions, the cassava tubers were incised manually with a sharp knife to a depth of about 4 mm up to the cortex of the cassava peels. The thread pitch of the spiral pattern and the distance of the meridian incisions were about 10 mm. Perforations were made with a needle in a 10 mm grid at a depth of about 4 mm.

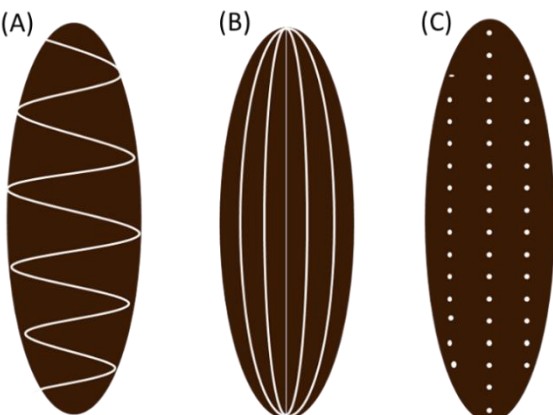

**Figure 4.** Incision patterns on cassava tubers for enzyme treatment. (**A**): spiral, (**B**): meridian, (**C**): perforation.

Based on our previous work [28], the incised cassava tubers were immersed in a Viscozyme® L solution at a dose of 1.2 mL g$^{-1}$ of cassava peels, a pH of 4.5, a temperature of 50 °C and an incubation time of 4 h. A blank solution, i.e., without adding the enzyme, was also used. As a control, incised cassava tubers were peeled without prior immersion. Mechanical peeling was performed using the above-mentioned peeling machine at a rotational speed of 850 rpm and a peeling time of 3 min. For each incision pattern, three experiments were conducted and the values were reported as mean ± SD.

### 3.2.3. Enzymatic Treatment

Prior to the mechanical peeling process, incised cassava tubers were immersed in an enzyme solution according to our previous work [28]. It was assumed in the present study that cassava peels account for 18% of the cassava tuber mass. The different dose of liquid Viscozyme® L (0.5, 1.2 and 1.9 mL $g^{-1}$ of cassava peels) was added to 5 L distilled water, and a container of the solution was placed on a heating plate with adjustable temperature settings and stirring function with a speed of 100 rpm (RT 15, IKA, Staufen im Breisgau, Germany). The pH was adjusted by 0.01 M HCl to the required pH. The enzyme solution was incubated at different times of 1.5, 4.0 and 6.5 h. A pH of 4.5 and a temperature of 50 °C of the immersion solution were chosen for all trials.

### 3.2.4. Peeling Process

After the enzymatic treatment, the treated cassava tubers were peeled using the prototype peeling machine by applying different peeling times (1.5, 3.0 and 4.5 min) and a speed of 850 rpm. For each parameter combination, three cassava tubers of similar shape and size were treated and peeled using the peeling machine.

### 3.2.5. Measuring the Peeling Effect

The efficiency of the peeling process was determined by peeled surface area and peel loss according to our previous work [42]. The peeled surface area was determined according to Srikaeo, Khamphu and Weerakul [43] by analyzing photos, which were taken from cassava tubers after the peeling process using an image processing software (Fiji, Madison, WI, USA). The peeled surface area, *PSA*, was calculated as

$$PSA = \frac{A_1}{A_2} \cdot 100, \tag{3}$$

where *PSA* is the peeled surface area after the peeling process (%), $A_1$ is the peeled surface area of the cassava tuber ($cm^2$), and $A_2$ is the surface area of the whole cassava tuber ($cm^2$).

The peel loss, *PL*, was calculated as

$$PL = \frac{m_1 - m_2}{m_1} \cdot 100, \tag{4}$$

where *PL* is the peel loss by the peeling process (%), $m_1$ is the mass of the unpeeled cassava tuber (g), and $m_2$ is the mass of the peeled cassava tuber (g).

### 3.3. Starch, Crude Fiber and Color of Cassava Peel Residues

Cassava peels are composed of water, starch, crude protein, crude fiber, minerals, cellulose, hemicellulose, lignin, sugars and phenolic compounds [44–46]. In this study, starch, crude fiber and color of cassava peel residues were measured as potential further indicators to evaluate the effect of the peeling process. Since the flesh of cassava tubers consists mainly of starch [5,11] and, in contrast, the peels have a high proportion of crude fiber (10.88%) [31] compared to the flesh (1.63%) [47], a high fiber/starch-ratio of the peel residues would indicate a low proportion of flesh and, hence, low losses. An influence on the color of the peel residues is also to be expected, as the outer skin is of dark color in contrast to the white flesh.

The starch content (SC) of the peel residue in % based on dry matter was determined according to Mitchell [48] using the polarimetric method [49]. The crude fiber (CF) of the peel residue in % based on dry matter was analyzed according to the van Soest method [50]. The color of the peel residue was determined in terms of CIELAB values (L*a*b*) using a colorimeter (CR-400, Konica Minolta, Japan). Only lightness (L) is presented in the study. The entire data set comprising L*a*b* values is provided in Table S1.

### 3.4. Box-Behnken Design

Response surface methodology (RSM) using a Box-Behnken design (BBD) was used to study the effect of enzymatic treatment with various enzyme doses and incubation times, peeling times and different sizes of tubers on the peeling process of cassava tubers. Based on the results of the pre-test, incision in a meridian pattern was chosen prior to immersion in the enzyme solution for the full set of trials.

The BBD consisting of 51 combinations with a three-level full factorial design was applied in this study. The selected independent variables were enzyme dose (0.5–1.9 mL g$^{-1}$ of cassava peels), incubation time (1.5–6.5 h), peeling time (1.5–4.5 min) and size of cassava tubers (small, medium and large). PSA and PL were chosen as the main responses to evaluate the cassava tuber peeling process. It was assumed that maximizing PSA towards 100% and PL towards 20% would lead to an optimum peeling process. The PL criteria was introduced to avoid losses by excessive peeling, considering the typical peel mass fraction of 15% to 20% in cassava [11,35].

### 3.5. Statistical Analysis

For the design of the experiments and the analysis of the data, Design-Expert 11 (STATCON GmbH, Witzenhausen, Germany) was used. The data was originally analyzed with the full polynomial model for each response. The full models were later adjusted by removing the statistically insignificant terms, not considering those required to support hierarchy. The final reduced polynomial models were then determined. The analysis of variance (ANOVA) was applied to examine the significance of independent variables and their interactions at *p* value < 0.05 (95% confidence level). The accuracy of the mathematical model was estimated using statistical analysis of coefficient of correlation ($R^2$) and mean absolute percentage error (*MAPE*). In order to validate the obtained optimal conditions of RSM for the peeling experiment with enzymatic treatment by the peeling machine, three replicates were conducted. The values were reported as mean ± SD.

## 4. Conclusions

The use of an enzyme treatment was found to be an effective approach to increase mechanical peeling efficiency. The most efficient operating conditions for an optimal peeling process were found at an enzyme dose of 0.8 mL g$^{-1}$, an incubation time of 4 h and a 3 min peeling time for small cassava tubers, an enzyme dose of 0.5 mL g$^{-1}$, an incubation time of 2.9 h and a 3.5 min peeling time for medium cassava tubers and an enzyme dose of 1.3 mL g$^{-1}$, an incubation time of 2.8 h and a 3.5 min peeling time for large cassava tubers. Under optimum conditions, a PSA of 82.03%, 80.90% and 89.52% was obtained for small, medium and large cassava tubers, respectively. The size of the tubers had a significant effect on the PSA of cassava tubers. However, there were no differences among the PL of different sizes of cassava tubers. Sorting the cassava tubers based on their size prior to mechanical peeling could be an effective way to improve the mechanical peeling process.

Correlations were established for fiber content, starch content and lightness of the peel residues and PL. It was determined that measuring the lightness of the peel residues has potential to develop a rapid measurement method to estimate the flesh loss during mechanical cassava peeling.

For practical application, the economic feasibility of the enzyme treatment should be taken into account. Therefore, in order to reduce the cost, recycling the enzyme from the enzymatic treatment process should be further studied. Furthermore, a mechanized device should be developed to make the meridian incisions on the tubers. It is also recommended that investigations are extended to further regions because of the wide variation in the properties of cassava tubers and different functional principles of locally used peeling machines.

**Supplementary Materials:** The following are available online at http://www.mdpi.com/2073-4344/10/1/66/s1, Figure S1: Normal probability plot of the residuals of peeled surface area, PSA and peel loss, PL (a&c) and plot of residuals versus predicted values of PSA and PL (b&d), Table S1: The experimental (Exp) and predicted (Pred)

values for crude fiber (CF), starch content (SC) and color parameters (L*a*b*) of cassava peel residue after enzyme treatment and mechanical peeling.

**Author Contributions:** Conceptualization, Z.B., S.L., S.R. and J.M.; methodology, Z.B. and S.R.; validation, Z.B.; formal analysis, Z.B.; investigation, Z.B.; resources, J.M.; data curation, Z.B.; writing—original draft preparation, Z.B.; writing—review and editing, S.L., S.R. and J.M.; visualization, Z.B.; supervision, S.L. and J.M.; project administration, S.L. and J.M.; funding acquisition, S.L. and J.M. All authors have read and agreed to the published version of the manuscript.

**Funding:** This research was funded by the German Federal Ministry of Education and Research (BMBF) under Project No. 031B0217 "CassavaUpgrade" and the foundation fiat panis (Ulm, Germany) under Project No. 33/2015 "Evaluation and optimization of enzymatic cassava roots peeling".

**Acknowledgments:** The authors would like to thank Jens Hartung for his suggestion for the improvement of the manuscript and Sabine Nugent for the English proofreading of this manuscript.

**Conflicts of Interest:** The authors declare no conflicts of interest.

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
