# Peer review of "Enzyme-Assisted Mechanical Peeling of Cassava Tubers"

_catalysts, doi:10.3390/catal10010066_

Round 1

Reviewer 1 Report

The originality of the study and the novelty it brings in the field is of actuality. The topic of manuscript entitled " Enzyme-assisted mechanical tubers peeling of cassava" is of interest because finding a way to increase the efficiency of mechanical peeling is very important.

The paper is well structured, the abstract is concise and in the topic; The purpose of the article and its significance is stated clearly; the introduction is supported by well selected bibliographic data.

The Experimental and Modeling Approach correctly. The manuscript is well written.

Results and Discussions could be improved by studying other papers in the field.

The conclusions are accurate and supported by the content. I advice to include some information regarding the advantages of the method used comparative with others techniques ( costs, time, etc.), this will increase the manuscript value.

Reviewer 2 Report

The paper from Barati et al. describes the use of enzymes pre-treatment to facilitate the peeling process of cassava tubers. Although it is not yet clear how economically feasible this peeling process would become in the future, this can be an interesting concept considering the growing importance of cassava tubers as global staple.

The introduction is clear and provides sufficient information to understand the purpose of the study. Likewise, the experimental design used to shape up the whole study was well thought and executed.

From a careful reading of the manuscript, the following issues have been identified which should be tackled before publication:

The author should ask a native speaker to review the English used in the manuscript, as some unclear expressions and typos have identified throughout the text When indicating the dose of Viscozyme® L to be added per weight unit of cassava peel, it is not clear how the weight of cassava peel is determined ‘a priori’. The authors should clear indicate this in the revised version of the manuscript. Concerning the relation between Peel Loss (PL), fiber content (CF) and starch content (SC), from the text in section 2.3 one might guess that CF+SC = PL. However, from the plots reported in Figure 3, one can clearly see that the CF+SC < PL. Do the authors have any idea of what the rest could be? This should be mentioned and possibly further explained in the text. This would be particularly relevant to the use of peels for further processes (such as cellulosic ethanol production). It would be good to have an additional set of control experiments where the efficiency of enzymes is investigated on tubers with no incision on their surface. In case this data is already available in previous publication, that should be clearly indicated in the text and properly referenced. Concerning that the rotation of the mechanical peeling was established at 850 rpm, can the authors explain whether this is a standard rotation frequency for that type of equipment or it can be varied? This can have an impact on the energy costs of a fully developed production process. In order to standardize this method, is there any quantitative relation between the total length of the incision on a tuber and the total surface of that tuber before peeling? This would be, of course, relevant in the case of meridian or spiral incisions. The cassava tubers used in this work come from Costa Rica. Do the authors think that there could be significant differences in the results from this study if tubers from different regions are used?
